# Noise Mapping, Prevalence and Risk Factors of Noise-Induced Hearing Loss among Workers at Muscat International Airport

**DOI:** 10.3390/ijerph19137952

**Published:** 2022-06-29

**Authors:** Norah A. Al-Harthy, Hassan Abugad, Najwa Zabeeri, Amal A. Alghamdi, Ghada F. Al Yousif, Magdy A. Darwish

**Affiliations:** 1IAU—Fellowship Program in Occupational Medicine, Department of Family and Community Medicine, College of Medicine, Imam Abdulrahman Bin Faisal University, Damam 34221, Saudi Arabia; 2Department of Family and Community Medicine, College of Medicine, Imam Abdulrahman Bin Faisal University, Damam 34221, Saudi Arabia; aaabugad@iau.edu.sa (H.A.); nazabeeri@iau.edu.sa (N.Z.); amlaghamdi@iau.edu.sa (A.A.A.); gfalyousif@iau.edu.sa (G.F.A.Y.); mdarwish@iau.edu.sa (M.A.D.)

**Keywords:** Oman, noise-induced hearing loss, noise exposure level, noise hazard, noise protection, Muscat International Airport, hearing protection devices

## Abstract

Noise-induced hearing loss (NIHL) is a common occupational hazard and a major cause of deafness among airport workers. However, few studies have been conducted to investigate the various risk factors related to hearing loss. Purpose: the purpose of this study was to measure the prevalence and risk factors of NIHL among Muscat International Airport airside workers. Method: Their daily noise exposure level at the airport was measured, and the time-weighted average (TWA) was calculated for each airside department. A cross-sectional study design involving 312 workers from the eight departments of the airport was chosen and the prevalence of NIHL among workers was assessed. The study participants then completed a self-administered questionnaire that covered their socio-demographic characteristics, occupational exposure history and the health-related risk factors of NIHL. Results: The TWA recorded for the workers was above the accepted limit in some departments, namely, cabin appearance, ramp, line maintenance and hangar. The prevalence of NIHL among participants was 21.79% (n = 68). Of these 68 participants with NIHL, 22.30% were exposed to job-related high noise levels. NIHL was common among participants aged 40 or above (57.35%, n = 39) and high school degree holders (29.60%, n = 29), as well as those who were exposed to higher noise levels (84.89%, n = 191) or who did not wear their hearing protection devices (HPDs) regularly (53.65%, n = 125). Conclusion: around a quarter of our study participants who were exposed to high noise levels suffer from NIHL.

## 1. Introduction

Noise-induced hearing loss (NIHL) is the second most common form of sensorineural hearing loss, after presbycusis (i.e., age-related hearing loss), and is one of the most common occupational diseases worldwide [1]. The World Health Organization (WHO) defines NIHL as “a permanent decrement in hearing threshold levels (HTLs), with a characteristic reduction of hearing sensitivity at the following frequencies: 3–4 kHz and/or 6 kHz if for more than 25 dB, and relatively better hearing sensitivity at nearby frequencies (i.e., 2 or 8 kHz) for 25 dB or less” [2].

Unfortunately, continuous and prolonged exposure to loud noise can kill nerve endings in the inner ear, resulting in permanent hearing loss that cannot be corrected via surgery or medication. Hence, this limits the ability to hear high frequency sounds and understand speech, which seriously impairs a person’s ability to communicate [3]. The damage to the inner ear cells depends on the intensity of sound and the duration of exposure to the noise; thus, according to the National Institute for Occupational Safety and Health (NIOSH), occupational noise exposure should not exceed the recommended exposure limit (REL) of 85 dB, as an 8 h time-weighted average (TWA) [4].

Several studies have been conducted to measure the prevalence of NIHL in several international airports. In a 2014 Korean study, the prevalence of NIHL was 49.4% [5]. In a Sudanese study in 2008, at Khartoum International Airport, the prevalence of NIHL was 55%, and in a 2018 Saudi Arabian study at King Khalid Airport it was 48% [6,7]. Noise mapping is an important tool for evaluating and interpreting environmental noise that can provide information to concerned authorities for the mitigation of noise pollution problems such as NIHL and inform about other risk factors for NIHL among manufacturing industry workers, including noise intensity, years of service, duration of exposure, age, hypertension, risk behaviors and hearing protection device (HPD) usage [8,9].

In Oman, Muscat International Airport has a newly opened airport building that was built over the last two years based on international standards to minimize noise levels airside [10]. However, no noise map was created to identify the magnitude of noise or the prevalence of NIHL among workers. This study aims to assess the noise exposure level among the airside workers at Muscat International Airport, as well as to measure the prevalence of NIHL and related risk factors. Noise within the airport airside environment includes not only the sounds of aircraft taxiing, starting up and landing, but also sounds from ground operations such as engine tests, run-up, auxiliary power units, baggage handling equipment, air conditioning, airside maintenance works, and traffic to and from the airport. This is why those who work airside are prone to NIHL.

## 2. Material and Methods

A cross-sectional study was conducted airside in the Muscat International Airport from July 2019 to August 2020 after obtaining both permission from the companies that worked there as part of the Oman Aviation Group and ethical approval from the Research and Ethical Review Board (MoH/CSR/19/10615) in Oman. The guidelines of the Declaration of Helsinki were used to inform the participants about the study; it was ensured that all information was in their language, including the purpose of the study, and that all information obtained in this study would be kept strictly confidential and anonymous. Airports are divided into landside and airside areas. The landside area is open to the public, whereas access to the airside area is tightly controlled. The airside is the part of an airport used by aircraft for loading, unloading and takeoffs, and includes runways, taxiways, the ramp, the storage hangar and aircraft maintenance facilities. The ramp is where aircraft are parked, unloaded or loaded, refueled, boarded or maintained. Line maintenance is any maintenance task that can be performed for the aircraft outside of a hangar in a short period, and is usually conducted on the ramp, whereas the hangar is where any longer aircraft maintenance is carried out. The maintenance department includes all maintenance work to ensure the safe operation of civil, mechanical, electrical and building equipment, infrastructure and facilities, and to ensure compliance with standards. Airfield operation workers coordinate between air-traffic control and maintenance personnel by dispatching, using airfield landing and navigational aids, implementing airfield safety procedures and monitoring. The cabin appearance department is responsible for cabin cleaning operations in aircrafts in ramp areas. The workshop department maintains equipment at the workshop, and lastly, there is the fire department which will respond to any incident and is usually located near the runway.

The study population was stratified to cover the eight departments at the airside of Muscat International Airport (fire, maintenance, airfield, hangar, line maintenance, ramp, cabin appearance and workshop).

This study was carried out in two steps. Firstly, workers’ daily exposure to noise from the airport airside was measured using a time-weighted average (TWA) sound level of 85 dB for 8 h per workday and a 3 dB exchange rate (i.e., for every 3 dB increase in noise level, the allowable exposure time was reduced by half), as this is the noise exposure limit recommended by the National Institute for Occupational Safety and Health (NIOSH) [11]. The TWA was measured using a personal noise dosimetry device (3M, ESP110315, USA), which was attached near the ear of chosen workers after the researcher explained to them how to wear it and how to use the device. For each department, the device was worn by at least two workers who had similar tasks in the same department, or by more than two workers who carried out different tasks within the same department (such as the ramp or maintenance department).

The maximum sound level (L_max_.), minimum sound level (L_min_.) and the level of the time-weighted average (L_TWA_) were taken at the end of each shift before calculating the mean of these measurements for several shifts. Based on TWA readings, the employees were divided into employees with a low noise exposure level (i.e., less than 85 dB for 8 h, or less than 83 dB for a shift of 12 h) and high-level exposure (85 dB or more in 8 h, or 83 dB or more for a shift of 12 h, as recommended by NIOSH [11]).

The second step was subdivided into two steps: second step A and second step B. In the second step A, data were collected from participants (n = 390) using a self-administered questionnaire, optionally validated by three otolaryngology (ORL) and occupational medicine consultants and divided into three sections. The first section was about a participant’s socio-demographic characteristics such as age (which was further categorized based on the distribution of the data into the following categories: under 30, 30–39 and 40+), nationality (Omani, non-Omani), gender, marital status (single, married) and education level (graduate level or higher (diploma, bachelor’s, master’s and PhD), high school degree and lower than high school degree (illiterate, elementary school and secondary school)). The second section was about the risk factors of NIHL, including family history of ear disease (yes, no), living near noisy areas (yes, no), current occupation location (fire, airfield, maintenance, workshop, hangar, line maintenance, ramp and cabin appearance), daily time spent on the job, years of working at the airport, chemical exposure (yes, no), hobbies (using noisy tools, riding motorcycles, gun shooting, attending discos/dances and playing musical instruments), smoking status (never smoked, current non-smoker (who stopped smoking before the study) and current smoker (who is still smoking during the study)), previous medical illnesses such as diabetes, hypertension and others (yes, no) and usage of ototoxic drugs such as aspirin, carbamazepine, amitriptyline and others (yes, no). The third section was about evaluation of noise risk at the workplace, including difficulty of communication due to noise (yes, no), wearing of hearing protective devices (HPDs; yes (at any time during work hours), no), time of wearing HPDs (frequently used, not frequently used) and available methods of HPDs for the current job (ear plugs, earmuffs, presence of noise isolation, regular machine maintenance, presence of reduced noise exposure period and training of workers on hazard).

In the second step B, a substudy was designed to screen for noise-induced hearing loss among the airside workers. This substudy sample was stratified to cover the eight departments at the airside of Muscat International Airport (fire, maintenance, airfield, hangar, line maintenance, ramp, cabin appearance and workshop). These airside workers were chosen randomly from a name list while considering shift times to cover all shifts and days of the week. The sample size was calculated using Epi Info software version 22, knowing that the total population size was n = 1948, assuming a 5% margin of error and 95% CI, and using a prevalence equal to that of the prevalence of NIHL at King Khalid International Airport in Saudi Arabia, which was found to be 48% [7]. Accordingly, the sample size was 312 participants (n = 312) after applying inclusion and exclusion criteria, as shown in Figure 1.

The excluded participants were those who had not yet completed 6 months in the airside area, workers with pre-employment ear disease or ear deformity, workers with previously diagnosed conductive hearing loss, workers with a history of a previous occupation with a high noise level exposure or those who were wearing medical hearing aids.

The prevalence of noise-induced hearing loss was determined based on the positive results of an audiogram of the pure audiometry test for more than 25 dB in high frequencies of 3–4 kHz and 6 kHz, interpreted by the researchers. The pure audiometry screening test has a sensitivity of 92% and specificity of 94% in detecting sensorineural hearing impairment [12] and in differentiating between NIHL and presbycusis (age-related hearing loss). The latter is characterized by bilateral hearing loss above 2000 hertz, and on a standard audiogram appears as an overall down-sloping line that represents impaired hearing at higher frequency sounds [13].

The analyses were performed using the SPSS version 22 software. The mean and standard deviation were calculated for the L_TWA_ for each shift per department. The chi-square test and Fisher’s exact test were used as appropriate to investigate the differences in the individual characteristics between the groups of our study participants. Univariate logistic regression models were used to estimate the risk of hearing loss in relation to several risk factors before these models were adjusted for age- and job-related noise exposure level. During the analyses, a *p*-value < 0.05 was considered significant.

## 3. Results

The airside workers’ noise exposure levels are summarized in Table 1, where the highest noise exposure level among the 12 h shift workers was reported among line maintenance department employees during their night shift (L_TWA_ = 84.7 dB, SD = 3.96, l_min_. = 60.70 l_max_. = 114.05), whereas the lowest was reported among airfield department employees during their night shift (L_TWA_ = 42.65, SD = 11.81, l_min_. = 61.50, l_max_. = 104.03). On the other hand, when considering eight-hour shift workers, the highest noise exposure level was reported for cabin appearance department employees during the night shift (L_TWA_ = 89.10, SD = 8.20, l_max_. = 117.05, l_min_. = 60.50), whereas the lowest noise exposure level was reported among the workshop department employees during their 8 h morning shift (L_TWA_ = 78 dB, SD = 2.97, l_max_. = 113.20, l_min_. = 60.50). Based on these reported average L_TWA_ measurements, which are presented in Table 1, the employees were divided into employees with high noise exposure levels who were working in the ramp, line maintenance, cabin appearance and hangar departments, and employees with low noise exposure levels who were working in the fire, airfield, maintenance and workshop departments.

The number of included participants was 312 out of 390 potential participants approached, with a response rate of 80%. Table 2 summarizes the socio-demographic features of the participants in relation to the noise levels at the airport airside. The majority of our participants were exposed to high noise levels (74.68%, n = 233), of whom 98.71 were males (n = 20), 87.12% Omanis (n = 203), 48.93% in their thirties (n = 114) and 42.92% were educated to high level than a high school diploma level (n = 100), as shown in Table 2.

Table 3 summarizes the distribution of job-related risk factors in relation to their noise levels at the airport airside. The majority of the participants were exposed to high noise levels (74.68%, n = 233), of whom 74.68% (n = 174) had a 12 h shift per day, 53.22% (n = 181) had less than 10 years of work experience, 81.97% wore hearing protection devices (n = 191) and 72.12% frequently wore HPDs (n = 225), as shown in Table 3.

### Noise-Induced Hearing Loss

Looking at the rate of NIHL among the participants, the audiometry test revealed that there were 68 cases, and of these 68 cases, 76.47% were exposed to high noise levels. Furthermore, 5.13% (n = 16) of the participants had a family history of hearing loss, 11.53% (n = 36) were current smokers, 9.29% (n = 29) lived near the airport, 16.99% (n = 53) practiced hobbies with possibly high noise levels (such as motorcycling, gun shooting and loud music), 12.82% (n = 40) had a medical condition that might affect their hearing and 7.37% (n = 23) used medication that might affect their hearing.

The mean work experience of our participants was 10.53 years (SD = 7.46) and NIHL was more common among participants with 10 years or less of work experience (n = 30, x2 (2) = (44.12), *p* < 0.004) and in participants older than 40 years old (x2 (2) = 57.35, *p* < 0.001). Additionally, NIHL was more common among participants who held a high school qualification (n = 29, x2 (2) = 42.65, *p* < 0.001), as shown in Table 4.

As shown in Table 5, it was found that the odds of developing NIHL were statistically significantly higher in participants aged 40 or older (OR = 7.07, 95% CI = 2.53 to 19.78) and in participants with over 20 years of work experience (OR = 3.38, 95% CI = 1.60 to 7.14). In contrast, the odds of developing NIHL were statistically significantly lower in participants with a higher academic qualification (OR = 0.35, 95% CI = 0.17 to 0.62).

## 4. Discussion

Measurements of noise exposure at Muscat International Airport showed that the mean L_TWA_ was above the accepted limit among the workers in the cabin appearance, ramp, line maintenance and hangar departments, which might be due to noise exposure in the working environment and peak exposures to noise, as well as the time spent at work.

Additionally, in this study some departments (i.e., airfield, fire, workshop and maintenance) reported a high peak of noise during some shifts (>100 dB), which might be sufficient to cause hearing loss as the workers need only 15 min of this level of noise to start developing NIHL based on the 3 dB exchange rate (i.e., for every 3 dB increase in noise level, the allowable exposure time was reduced by half that recommended by the National Institute of Occupational Safety and Health (NIOSH) [2]). The study showed similar noise exposure that was measured while working in a hangar was between 70 and 91 dB(A), and when following the aircraft on the station ramp it was 81 dB(A) during an assumed 8 h workday [14]. However, peak exposures could be very high, and the highest exposure levels occurred during sheet-metal work and while using riveting hammers in hangars [14]. Unfortunately, the HPDs (such as earmuffs and ear plugs) can only reduce noise within the range of the noise reduction rate (NRR) (22–30 dB) if they are worn correctly [15]; hence, some departmental workers remain under risk of NIHL regardless of their HPD protection measures.

The prevalence of NIHL among the total study population of workers was 21.79%, whereas it was 5.13% in employees with low noise exposure and 16.67% in employees with high noise exposure. Comparing these results with the reported prevalence of NIHL in the literature, which was around 33.5–49.4%, the reported NIHL prevalence in this study was low [5,16]. This difference in the measured prevalence might have resulted from the characteristics of the included participants. For example, the overall prevalence of NIHL in the Jomo Kenyatta International Airport was 15.3%, but the target population included both aircrew and ground crew, whereas the population in this study only included ground crew [17]. It might be important to do periodic audiometry screening and training to enhance good practices for Muscat International Airport employees in order to detect and/or prevent any future hearing loss [5]. The use of engineering controls is prioritized over administrative controls and personal protective equipment, and that is what was seen in the fire department at Muscat Airport as they have insulation in the fire department building to minimize noise exposure.

Studies have revealed that several factors could contribute to the development of hearing loss besides high noise exposure levels, such as aging and prolonged working experience [16]. Nasir and Rampal claimed that older age (especially >45 years) and working in a noisy environment for longer than 5 years were two important risk factors that might cause permanent hearing loss [16]. However, at Muscat International Airport most of the participants were in their thirties and had few years of working experience (<5 years). Other studies reported that the greatest hearing loss occurred during the first 10–15 years of working in a noisy environment,^16^ which emphasizes the importance of frequent hearing screening for workers to detect and prevent future permanent nerve damage due to noise.

Other factors that may affect the severity of NIHL and elevate its risk in an occupationally noise-exposed worker is the presence of chronic diseases such as diabetes, as well as smoking—especially if smoking more than 20 packs per year [18,19]. In this study, 8.82% of the participants who smoked also had NIHL. However, in the literature a higher rate of NIHL was found amongst smokers and was reported to reach 55% of included smoker participants in some studies [19]. In this study, 25% of the participants had chronic diseases and 8% were taking medication; thus, regular general medical checkups might be recommended to prevent the augmentation of hearing loss in predisposed workers such as those who are older or have had more years of noise exposure.

In our study, most airside airport workers infrequently wore HPDs (such as earmuffs and ear plugs). Studies have shown that infrequent usage of HPDs with a high occupational level of noise exposure might be a contributing factor for hearing loss [19].

In this regard, it was recommended by the Occupational Safety and Health Administration (OSHA) that airport employers should provide HPDs to all their employees, who must use them whenever noise exposure meets or exceeds 85 dB [19]. In addition, OSHA instructed employers to implement a hearing conservation program when noise exposure was at or above 85 dB, averaged over 8 working hours or an 8 h TWA [20]. Unfortunately, there was no hearing conservative program at the airport, although instructions from OSHA regarding those areas with a high risk of noise exposure had already been received; therefore, this is one of the top recommendations for the airport from this study, as conducting a program could help prevent initial occupational hearing loss, and preserve and protect remaining hearing [21].

### Study Limitations

Using a cross-sectional design did not allow us to establish a causality relationship in addition to errors that arose from recall bias, especially in relation to the magnitude and frequency of noise exposure.

## 5. Conclusions

In conclusion, although the airport is well established and has the highest standards of noise reduction measures, the noise exposure was just above the accepted level in some departments and included some very high peak exposures in others, which is relevant to our results concerning NIHL. In addition to noise exposure level as a predictor for hearing loss, NIHL was found to be related to older age, lower educational achievements, fewer years of experience and irregular wearing of hearing protection. This study highlighted the importance of further preventive measures, including regular audiometric screening for early detection of hearing loss and ongoing hearing protection conservation programs. In addition, the study demonstrated the importance of administrative controls and engineering controls, such as having SOPs (standard operating procedures) to minimize work hazards in the workplace, employee training and education, frequent airport machine maintenance, regular inspection, recording and supervision, and advisors emphasizing to their workers that they must always wear their HPD in the proper manner during any noise exposure.

## Figures and Tables

**Figure 1 ijerph-19-07952-f001:**
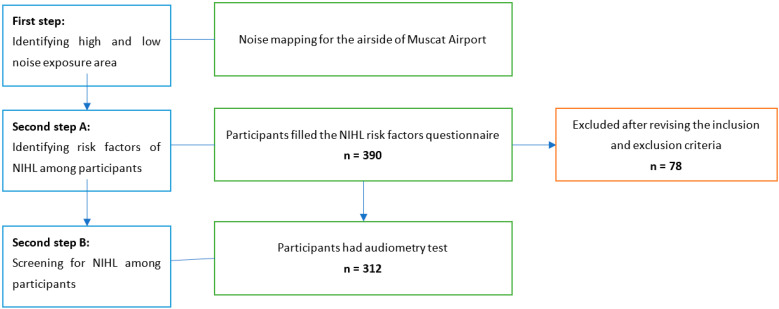
Flow chart showing the two steps that were carried out in this study. The total number of participants was 390, however, we only included n = 312 in the analysis, as seen in the diagram.

**Table 1 ijerph-19-07952-t001:** Average noise exposure level (L_TWA_) measured in dB for airport departments by shift.

Department	Fire M (SD) (n = 24)	Airfield M (SD) (n= 18)	Maintenance M (SD) (n = 19)	Line Maintenance M (SD) (n= 47)	Hangar M (SD) n = (21)	Work Shop M (SD) n= (18)	Cabin Appearance M (SD) n = (13)	Ramp M (SD) n = (152)
Morning Shift	75.10 dB (2.66)	69.18 dB (13.10)	59.16 dB (14.07)	82.40 dB (5.37)	83.05 dB (2.76)	78.00 dB (2.97)	82.70 dB (5.10)	79.25 dB (8.57)
Afternoon Shift	No scheduled shift	No scheduled shift	No scheduled shift	No scheduled shift	No scheduled shift	No scheduled shift	82.50 dB (4.27)	86.1 dB (5.42)
Night Shift	75.88 dB (4.87)	42.65 dB (11.81)	69.08 dB (9.51)	84.70 dB (3.96)	65.70 dB (13.47)	No scheduled shift	89.10 dB (8.20)	78.63 dB (7.32)

M = mean; SD = standard deviation.

**Table 2 ijerph-19-07952-t002:** Distribution of study participants’ socio-demographic features in relation to their noise exposure level as identified using l_TWA_ measurements.

	Employees with Low Noise Exposure Level		Employees with High Noise Exposure Level		Total	Chi-Square Test		
	n = 79	(25.32%)	n = 233	(74.68%)	n = 312	X^2^	df	*p*
**Age**						15.208	2	<0.001
≤30 years	33	(41.77)	46	(19.74)	79 (25.32)			
31–39 years	27	(34.18)	114	(48.93)	141 (45.19)			
≥40 years	19	(24.05)	73	(31.33)	92 (29.49)			
**Nationality**						1.144	1	0.285
Omani	65	(82.28)	203	(87.12)	268 (85.90)			
Non-Omani	14	(17.72)	30	(12.88)	44 (14.10)			
**Gender**								
Male	79	(100)	203	(98.71)	309(99.04)	1.027	1	* 0.574
Female	0	0	3	(1.29)	3(0.96)			
Married status						1.036	1	0.309
Single	(61)	(77.22)	192	(82.40)	253(81.09)			
Married	18	(22.78)	41	(17.60)	59(18.91)			
**Educational level**								* <0.001
Less than high school	2	(2.53)	46	(19.74)	48 (15.38)			
High school degree	11	(13.92)	87	(37.34)	98(31.41)			
Higher than high school	66	(83.55)	100	(42.92)	166 (53.21)			

df = degree of freedom, *p* = *p*-value. * indicates that *p*-value came from Fisher’s exact test.

**Table 3 ijerph-19-07952-t003:** The distribution of job-related risk factors in relation to noise exposure level.

	Employees with a Low Noise Exposure Level		Employees with a High Noise Exposure Level		Total	Chi-Square Test		
	n = 79	(25.32%)	n = 233	(74.68%)	n = 312	X^2^	df	*p*
**Length of shift per day**						6.584	1	0.010
Less than 12 h	32	(40.51)	59	(25.32)	91 (29.17)			
12 h	47	(59.49)	174	(74.68)	221 (70.83)			
**Work experience**								
≤10 years	57	72.15	124	53.22	181 (58.01)			*0.03
11–20 years	15	18.99	73	31.33	90 (28.85)			
21–30 years	3	3.80	35	15.02	38 (12.18)			
>30 years	1	1.27	2	0.86	3 (0.96)			
**Chemical exposure**							1	0.378
Yes	29	(36.71)	73	(31.33)	102 (32.69)			
No	50	(63.29)	160	(68.67)	210 (67.31)			
**Wearing hearing protection devices (HPDs)**								
Yes	34	(43.04)	191	(81.97)	225 (72.12)	44.478	1	<0.001
No	45	(56.96)	42	(18.03)	87 (27.88)			
**HPD was not available**								
Yes	17	(21.52)	12	(5.15)	29 (9.29)	18.749	1	<0.001
No	62	(78.48)	221	(94.85)	283 (90.71)			
**Use of ear plugs**								
Yes	23	(29.11)	130	(55.79)	153 (49.04)	16.804	1	<0.001
No	56	(70.89)	103	(44.21)	159 (50.96)			
**Use of earmuffs**								
Yes	42	(53.16)	126	(54.08)	168 (53.85)	0.020	1	0.888
No	37	(46.84)	107	(45.92)	144 (46.15)			
Frequent usage of HPD								
Yes	34	(43.04)	191	(81.97)	225 (72.12)	44.478	1	0.000
No	45	(56.96)	42	(18.03)	87 (27.88)			
**Isolation from noise**								
Yes	8	(10.13)	11	(4.72)	19 (6.09)	3.014	1	0.083
No	71	(89.87)	222	(95.28)	293 (93.91)			
**Regular machine maintenance**								
Yes	2	(2.53)	4	(1.72)	6 (1.92)			* 0.645
No	77	(97.47)	229	(98.28)	306 (98.8)			
**Administrative plan to minimize time of exposure**								
Yes	23	(29.11)	63	(27.04)	86 (27.56)	0.127	1	0.721
No	56	(70.89)	170	(72.96)	226 (72.44)			
**Received training to protect them from noise hazard**								
Yes	10	(12.66)	25	(10.73)	35 (11.22)	0.220	1	0.639
No	69	(87.34)	208	(89.27)	277 (88.78)			

* *p*-value from Fisher’s exact test.

**Table 4 ijerph-19-07952-t004:** Distribution of individual characteristics in relation to audiometry test results.

	Positive Audiometry Test (NIHL) 68 (21.79%)	Normal Audiometry 244 (78.21%)	Total 312	Chi-Square Test		
				X2	Df	*p*
**Age**						
<30	8 (11.77)	71 (29.11)	79 (25.32)	33.147	2	0.000
31–39	21 (30.88)	120 (49.18)	141 (45.19)			
40+	39 (57.35)	53 (21.71)	92 (29.49)			
**Educational level**						
High school degree	29 (42.65)	69 (28.28)	98 (31.41)	21.685	2	0.000
Higher than high school degree	20 (29.41)	146 (59.84)	166 (53.21)			
Less than high school degree	19 (27.94)	29 (11.88)	48 (15.38)			
**Smoking**						
Never smoked	50 (73.53)	176 (72.13)	226 (72.44)	0.737	2	0.692
Current non-smoker	12 (17.65)	38 (15.57)	50 (16.03)			
Current smoker	6 (8.82)	30 (12.30)	36 (11.53)			
**Medical illness**						
No	58 (85.29)	214 (87.70)	272 (87.18)	1.123	1	0.289
Yes	10 (14.71)	30 (12.30)	40 (12.82)			
**Medication**				2.393	1	0.122
No	60 (83.82)	229 (93.85)	289 (92.63)			
Yes	8 (11.76)	15 (6.15)	23 (7.37)			
**Family history of hearing loss**						
No	63 (21.3)	233 (78.7)	296 (94.87)	0.885	1	0.347
Yes	5 (31.3)	11 (68.8)	16 (5.13)			
**Living near noisy area**						
No	66 (97.06)	217 (88.93)	283 (90.71)	4.194	1	* 0.056
Yes	2 (2.94)	27 (11.07)	29 (9.29)			
**Working with high noise levels**						
No	16 (20.3)63	63 (79.7)	79	0.148	1	0.701
Yes	52 (22.3)	181 (77.7)	233			
**Work experience in airport**						
1–10	30 (44.12)	151 (61.89)	181 (58.01)	11.183	2	0.004
11–20	22 (32.35)	68 (27.87)	90 (28.85)			
>20 years	16 (23.53)	25 (10.24)	41 (13.14)			
**Chemical exposure at work**						
No	50 (73.53)	160 (65.57)	210 (67.31)	0.939	1	0.333
Yes	18 (26.47)	84 (34.43)	102 (32.69)			
**Specific hobbies exposed to noise**						
No	61 (89.71)	198 (81.15)	259 (83.01)	2.493	1	0.114
Yes	7 (10.29)	46 (18.85)	53 (16.99)			
**Wearing HPD**						
Yes	56 (24.9)	169 (75.1)	225	4.532	1	0.033
No	12 (13.8)	75 (78.2)	87			
**Use of ear plugs**						
Yes	35 (22.9)	118 (77.1)	153	0.206	1	0.650
No	33 (20.8)	126 (79.2)	159			
**Use of earmuffs**						
Yes	38 (20)	130 (77.4)	168	0.145	1	0.703
No	30 (20.8)	114 (79.2)	144			

**Table 5 ijerph-19-07952-t005:** Calculated odds ratios that estimate the risk of developing NIHL in relation to several risk factors.

	Unadjusted OR	95% CI	Adjusted OR	95% CI
**Age *^,a^**				
<30	Reference			
31–39	0.15	0.07, 0.36	1.62	0.65, 4.04
40+	0.24	0.13, 0.44	7.07	2.53, 19.78
**Educational level *^,a^**				
High school degree	Reference			
Higher than high school degree	0.35	0.17, 0.62	0.30	0.15, 0.60
Less than high school degree	1.56	0.76, 3.21	1.40	0.66, 2.96
**Smoking ^a^**				
Never smoked	Reference			
Current non-smoker	1.24	0.48, 3.20	1.50	0.70, 3.24
Current smoker	1.58	0.53, 4.70	1.09	0.41, 2.89
**Medical illness ^a^**				
No	Reference			
Yes	1.54	0.69, 3.41	1.06	0.45, 2.50
**Medication ^a^**				
No	Reference			
Yes	1.08	0.94, 1.23	1.05	0.92, 1.21
**Family history of hearing loss ^a^**				
No	Reference			
Yes	1.68	0.56, 5.02	0.57	0.18, 1.76
**Living near noisy area ^a^**				
No	Reference			
Yes	0.24	0.07, 1.05	4.25	0.96, 18.70
**Working with high noise levels ^b^**				
No	Reference			
Yes	0.88	0.47, 1.66	1.05	0.54, 2.05
**Work experience in airport *^c^**				
1–10	Reference			
11–20	1.71	0.91, 3.20	0.93	0.46, 1.88
>20 years	3.38	1.60, 7.14	0.79	0.29, 2.14
**Chemical exposure at work ^a^**				
No	Reference			
Yes	0.69	0.38, 1.25	1.51	0.82, 2.78
**Specific hobbies exposed to noise ^a^**				
No	Reference			
Yes	0.51	0.22, 1.19	0.53	0.22, 1.26
**Wearing HPD *^,a^**				
Yes	Reference			
No	2.07	1.05, 4.09	2.07	0.99, 4.34
**Use of ear plugs ^a^**				
Yes	Reference			
No	0.62	0.52, 1.51	0.89	0.51, 1.56
**Use of earmuffs ^a^**				
Yes	Reference			
No	0.90	0.52, 1.55	0.93	0.54, 1.62

*^,a^ Adjusted regression models for working with high noise levels and years of experience; ^b^ adjusted regression models for years of experience; ^c^ adjusted regression models for age and working in an area with high noise levels; bold font indicates significant *p*-value.

## Data Availability

The data presented in this study is available on request from the corresponding author. The data is not publicly available due to data privacy.

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
