# Peer review of "Noise Mapping, Prevalence and Risk Factors of Noise-Induced Hearing Loss among Workers at Muscat International Airport"

_ijerph, 2022, doi:10.3390/ijerph19137952_

Round 1
Reviewer 1 Report
See attached file

Author Response
Thank you for your comments and corporation .
We really appreciate your points and we modified the point that need to be change .
kindly see the attchement
Thank you

Reviewer 2 Report
Type of manuscript: Article
Title: Noise Mapping of Muscat International Airport and Noise Induced Hearing Loss among its Workers
Comments and Suggestions for Authors
Title
Rephrase the title for better understanding on the concept of the whole manuscript
Abstract
Line 11: to add one or two sentences for intro
Line 11: Purpose – risks factors or risk factors?
Line 20: Results – some departments … to specify which department
Introduction
Only explained about NIHL
How about noise mapping and risk factors?
Line 59: study aim only mentioned about prevalence of NIHL
Material and methods
Line 63: the info was break up into two lines
Line 73: what are the job scopes for each selected departments in this study?
Line 114: why author’s name was stated here?
Line 116: do you still used ENT and not ORL
Better to draw in consort diagram to show how the three steps were carried out in this study
Self-administered questionnaire was the last step?
When were the study criteria (inclusion and exclusion) applied? During second step or before the first step started?
Line 126: what was the operational definition for currently non-smoker and currently smoker?
Line 129: for third section – any specific instructions given to participants to answer? For example, wearing of hearing protective devices (HPD; yes, no) if the worker just wore only for five minutes do we consider YES?
Results
Table 1: to put in the methodology the total numbers of worker for each department so that we know the proportion (n in Table 1)
Line 158: why put sub study? Prevalence of NIHL was already part of this research
Table 2: why age was categorized as such? Can fisher exact test been applied for education level (3x2 table)?
Line 167: X2= chis-square test?
Line 168: fisher exact test929 … 929 referring to what?
Line 170: what do you mean by possible cases? Was it non confirmatory?
Table 3: can you explained how fishers exact test can be applied in work experience because many cells less than 5. Was the assumptions meet?
Table 4:
what do you mean by positive audiometry test?
please put p-value
why there were two values for chi-square ie 0.05 and 0.01?
better to have one more table for final model logistic regression
Discussions
Line 202 till 208: to put under introductions
Line 209 till 211: please discuss, not just repeat the results
Line 212 and 213: unsupported statement
Please discuss for these results – with reasons:
however, of these 68 possible cases, 22.30% were exposed to high noise level … Line 170-171 why only 22%? Supposed more than that
line 176: more common among participants with ≤ 10 years of work experience (n=28)
line 188: NIHL was more common among participants older than 40 years old 188 (n=39) how to differentiate with presbycusis?
Line 265 till 268: OSHA already instructed to implement hearing conservation program so how the achievement? Was it good especially to this airport? High risk area should have the best SOP (standard operating procedure) to minimize work hazards from workplace
Study limitations
Line 273 till 276: why it was put under limitation? I thought it was plan as such to minimize the bias
Conclusion
These conclusions are more like a re-statement of results and recommendations. The conclusion paragraph should restate the thesis, summarize the key supporting ideas discussed throughout the work, and offer your final impression on the central idea
Suggested Revisions:
Need revisions before this manuscript can be published
Author Response
Thank you for your comments and corporation .
We really appreciate your points and we modified the point that need to be change and we tried our best to clarify each point.
kindly see the attachment
Thank you

Reviewer 3 Report
The paper describes noise levels measured at Muscat International airport in Oman, and the prevalence of NIHL, together with associated risk factors. The paper is interesting and relevant, but needs extensive revision in terms of grammar.
Abstract
1. Several phrases are unnecessarily repeated.
2. It is not necessary to present the X2 values in the results (both here and in the body of the paper).
Introduction
1. The paragraph starting "Unfortunately, NHLS was found..." (lines 49-51) does not fit well here.
2. more detail is required in the paragraph starting "Several studies..." (lines 52-55). For example, in what year was the prevalence of NIHL in Korea 49.4%?
Materials and methods
1. The second and third paragraph contain some aspects about ethics. Ethical issues should be described together in a single paragraph.
2. The authors say that the study was "carried out in three steps", yet only two steps are described.
3. The selection of the study participants were selected for the noise measurement step.
4. Lines 91-95 relate to data management. It may be that the data management and analysis for noise measurements is described before the second step, but I see nothing about data analysis in this section.
5. It is not clear if the participants in the second step are the same as those in the first step. In the results section, Table 2 seems to relate to NIHL, as indicated in the text (line 160), but the column headings are "noise exposure level". Where do these data come from - the first or second step of the study?
Results
1. The main findings presented in the tables are not adequately described in the text.
2. Table 1: units need to be added (dB). Does M in the column headings stand for 'mean'?
3. Table 2: lines 158-164 relate to Table 2, and males are stated to predominate, yet sex is not included in the table. Also see comment 5 under "Methods' above.
4. Table 2: it is not necessary to present the X2 values of the degrees of freedom. There are spelling errors in the footnote.
5. Table 2: column percentages should be calculated, rather than row percentages.
6. Lines 173: what are the units?
7. Table 3: as for Table 2 (comments no. 4 and 5)
8. Table 3: why is 'Methods' written before "HPD was not available"?
9. Lines 188 - 189: refer to Table 4 here, rather than at the end of the paragraph, to guide the reader.
10. The title of Table 4 should be revised - no need to mention "Summary of regression analysis models". The table is difficult to read in portrait - rather use landscape.
11. Table 4: consider including p values (this is conventional in regression tables). Also see comment no. 5 above.
Discussion
1. Section 4.1: why is the 2nd point a limitation (lines 273-276)?
Other
1. Don't use first names in the author contributions
Author Response
Thank you for your comments and corporation .
We really appreciate your points and we modified the point that need to be change and we tried our best to clarify each point .
kindly see the attchment
Thank you

Round 2
Reviewer 2 Report
Satisfy with the correction done